# In Silico Identification and Validation of Pyroptosis-Related Genes in Chlamydia Respiratory Infection

**DOI:** 10.3390/ijms241713570

**Published:** 2023-09-01

**Authors:** Ruoyuan Sun, Wenjing Zheng, Shuaini Yang, Jiajia Zeng, Yuqing Tuo, Lu Tan, Hong Zhang, Hong Bai

**Affiliations:** Key Laboratory of Immune Microenvironment and Disease (Ministry of Education), Department of Immunology, School of Basic Medical Sciences, Tianjin Medical University, Tianjin 300070, China; sunry0609@163.com (R.S.); joeyzheng1986@126.com (W.Z.); nier1998@163.com (S.Y.); zjiajia814@163.com (J.Z.); tuoyuqing2021@163.com (Y.T.); tanlu@tmu.edu.cn (L.T.); zhanghong0621@tmu.edu.cn (H.Z.)

**Keywords:** pyroptosis, *Chlamydia trachomatis* infection, inflammatory death, competitive endogenous RNA regulatory network, immune infiltration

## Abstract

The incidence of *Chlamydia trachomatis* respiratory infection is increasing, and its pathogenesis is still unclear. Pyroptosis, as a mode of inflammatory cell death, plays a vital role in the occurrence and development of *Chlamydia trachomatis* respiratory infection. In this study, the potential pyroptosis-related genes involved in *Chlamydia trachomatis* respiratory infection were identified by constructing a mouse model of *C. muridarum* infection combined with bioinformatics analysis. Through in-depth analysis of the RNA sequencing data, 13 differentially expressed pyroptosis-related genes were screened, including 1 downregulated gene and 12 upregulated genes. Gene ontology (GO) analysis showed that these genes mainly regulate inflammatory responses and produce IL-1β. Protein–protein interaction network analysis identified eight hub genes of interest: *Tnf*, *Tlr2*, *Il1b*, *Nlrp3*, *Tlr9*, *Mefv*, *Zbp1* and *Tnfaip3*. Through quantitative real-time PCR (qPCR) analysis, we found that the expression of these genes in the lungs of *C. muridarum*-infected mice was significantly reduced, consistent with the bioinformatics results. At the same time, we detected elevated levels of caspase-3, gasdermin D and gasdermin E proteins in the lungs of *C. muridarum*-infected mice, demonstrating that *Chlamydia trachomatis* infection does induce pyroptosis. We then predicted nine miRNAs targeting these hub genes and constructed a key competitive endogenous RNA (ceRNA) network. In summary, we identified six key pyroptosis-related genes involved in *Chlamydia trachomatis* respiratory infection and constructed a ceRNA network associated with these genes. These findings will improve understanding of the molecular mechanisms underlying pyroptosis in *Chlamydia trachomatis* respiratory infections.

## 1. Introduction

Chlamydia is a genus of obligate intracellular bacteria, including four chlamydia species, *Chlamydia trachomatis*, *Chlamydia pneumoniae*, *Chlamydia psittaci* and *Chlamydia pecorum*, among which *Chlamydia trachomatis* and *Chlamydia pneumonia* are most closely associated with human health [1,2]. *Chlamydia trachomatis* infection can lead to trachoma and sexually transmitted diseases [3]. Still, the incidence rate of lung infections caused by *Chlamydia trachomatis* is high clinically, and the age of onset is getting younger and younger. As one grows older, there may also be recurrent infections of *Chlamydia trachomatis* [4]. Therefore, elucidating the pathogenesis of *Chlamydia trachomatis* respiratory tract infection and providing treatment according to it may obtain better therapeutic effects [5]. To design better strategies to control chlamydia respiratory tract infection and understand the complex heterogeneity of this disease at unbiased transcriptome depth, we used *Chlamydia muridarum* (*C. muridarum*), the first strain isolated from mouse lung tissue, to construct a mouse model of *C. muridarum* respiratory tract infection and performed RNA sequencing on lung tissues from healthy and *C. muridarum*-infected mice to investigate the effects of *C. muridarum* infection on lung inflammation [6] and immune cell infiltration, and help understand the dynamics of the immune response at the molecular level and cellular level.

Pyroptosis is a novel form of programmed cell death mediated by gasdermin family proteins [7], which is distinct from other modes of cell death, such as apoptosis and cell necrosis, and is characterized by the dependence on inflammatory caspase and the release of large numbers of proinflammatory factors [8]. According to the activation mechanism, it can be divided into caspase-1-dependent and caspase-1-independent pathways [9]. The activation of caspase-1 characterizes the canonical pathway. It can cleave the N-terminus of gasdermin D (GSDMD) and bind to phospholipid proteins on the cell membrane to form pores, which subsequently cause cytoplasmic swelling and release of contents. In addition, activated caspase-1 promotes the formation of active IL-1β and IL-18, which are released to the extracellular space and cause inflammation [10,11]. The noncanonical pathway depends on caspase-4/5/11 and other caspase family members, which also cleave GSDMD after activation, leading to pyroptosis [12,13]. Under physiological conditions, pyroptosis is essential for the maintenance of innate immunity and the prevention of tumorigenesis. Still, accumulated data have suggested that abnormal pyroptosis is associated with cardiovascular diseases [14], nervous system diseases [15] and pulmonary diseases such as chronic obstructive pulmonary disease and asthma [16,17]. However, there is little information concerning pyroptosis in *Chlamydia trachomatis* respiratory infections. Further study on the role of pyroptosis in lung tissue cells in *Chlamydia trachomatis* respiratory tract infection and its molecular mechanism will provide new ideas for the pathogenesis of chlamydia respiratory tract infection.

In this study, we explored potential pyroptosis-related genes by analyzing gene expression in the lung tissue of mice infected with *C. muridarum*. Through correlation analysis, gene ontology (GO) enrichment analysis, Kyoto Encyclopedia of Genes and Genomes (KEGG) pathway analysis and protein–protein interaction (PPI) network analysis, we identified pyroptosis-related genes involved in *C. muridarum* infection. We validated them by quantitative real-time PCR (qPCR). Then, we predicted the correlated targeting of microRNAs (miRNAs) and long noncoding RNAs (lncRNAs) associated with key pyroptosis genes and constructed a competitive endogenous RNA (ceRNA) regulatory network. This lays the foundation for revealing the mechanism of pyroptosis in pyroptosis infection and generating therapeutic insights (Figure 1).

## 2. Results

### 2.1. Screening of 13 Differentially Expressed Pyroptosis-Related Genes

To investigate the effect of *Chlamydia trachomatis* infection on lung tissue inflammation, we established a mouse model of *C. muridarum* respiratory infection, and transcriptome sequencing was performed on the lung tissues of mice after a 7-day infection. The results of RNA sequencing before and after standardization are shown in Figure 2A. To assess the reproducibility of the data, we performed a principal component analysis of the data (Figure 2B). Using *p*-value < 0.05 and |log2 foldchange| > 2 as thresholds, 539 differentially expressed genes (including 358 upregulated genes and 181 downregulated genes) were obtained. The intersection of 539 differentially expressed genes and 184 pyroptosis-related genes collected in the GEO database identified 13 differentially expressed pyroptosis-related genes, which are marked on the volcano map (Figure 2C,D). The expression of these genes was visualized using a heatmap and box plot (Figure 2E,F). We also found strong correlations among the 13 differentially expressed pyroptosis-related genes (Figure 2G). In this part, we obtained 1 downregulated pyroptosis-related gene, *Adipoq,* and 12 upregulated pyroptosis-related genes, *Trem2*, *Aoah*, *Tlr2*, *Ccr5*, *Tnfaip3*, *Tlr9*, *Tnf*, *Zbp1*, *Nlrp3*, *Clec5a*, *Il1b* and *Mefv*.

### 2.2. Functional and Pathway Enrichment Analysis of 13 Differentially Expressed Pyroptosis-Related Genes

To further understand the potential molecular mechanisms and signaling pathways of differentially expressed pyroptosis-related genes, we conducted an enrichment analysis of them. The results of the GO analysis showed that 496 GO terms were significantly enriched, including 487 biological processes (BP), 3 cellular components (CC) and 6 molecular functions (MF), mainly involved in the regulation of inflammatory response, inflammasome complex and interleukin-1 receptor binding, which suggests that these pyroptosis-related genes play a role in *C. muridarum* infection through pyroptosis and inflammatory response (Figure 3A–C). We analyzed the relationships between the pathways and analyzed the shared genes of the first five pathways to explore the relationships between differentially expressed pyroptosis-related genes and these pathways (Figure 3D,E). At the same time, KEGG analysis revealed 55 significant pathways; the results showed that differentially expressed proptosis-related genes play a key role in necroptosis, the NOD-like receptor signaling pathway and other pathways (Figure 4). The most significant paths obtained from GO and KEGG analyses are shown in Table 1. Both GO and KEGG analysis indicated that 13 differentially expressed pyroptosis-related genes were involved in *C. muridarum* respiratory infection, mainly through regulating inflammatory response and the NOD-like receptor signaling pathway.

### 2.3. Construction of PPI Network of 13 Differentially Expressed Pyroptosis-Related Genes

The PPI network is constructed according to the interaction between each protein, which is of great significance for understanding the function and interaction of proteins. Firstly, network files selected by R software (version 4.1.3) were uploaded to the STRING database, an online tool for exploring known and predicted protein–protein interactions. A PPI network with 13 nodes and 40 edges was obtained, with an average node depth of 6.15 (Figure 5A). We then used the cytoHubba plug-in of the Cytoscape software for further analysis, using the MCC algorithm to rank differentially expressed pyroptosis-related genes and screen out the top eight hub genes, namely, *Tnf, Tlr2, Il1b, Nlrp3, Tlr9, Mefv, Tnfaip3* and *Zbp1* (Figure 5B, Table 2).

### 2.4. Validation of Pyroptosis-Related Genes

To ensure the reliability of the RNA sequencing data analysis results, we further detected the expression levels of the eight key differentially expressed pyroptosis-related genes with qPCR. The results showed that the mRNA expression levels of these eight genes were significantly increased in the lung tissues of *C. muridarum*-infected mice (Figure 6). Meanwhile, the Western blot showed that caspase-3, GSDMD and GSDME protein levels were significantly increased (Figure 7). These results indicate that the eight key differentially expressed pyroptosis-related genes are upregulated considerably after *C. muridarum* infection, consistent with the results of bioinformatics analysis, and *C. muridarum* infection does lead to pyroptosis in lung tissue.

### 2.5. Construction of ceRNA Regulatory Network for Key Pyroptosis-Related Genes

To further explore the relationship between lncRNA and miRNA in *C. muridarum* infection and their role in regulating the expression of pyroptosis-related genes, we constructed a ceRNA regulatory network. MultiMiR is a new miRNA-target interaction R package and database that combines human and mouse records from 14 databases and includes the mirTarbase database and experimental methods for validating interactions. We used luciferase reporter gene analysis as a screening index to predict nine miRNAs, including mmu-miR-125b-5p, mmu-miR-29b-3p, etc. (Table 3). Then, based on these miRNAs, 27 targeted lncRNAs were obtained through the Starbase database. Finally, the ceRNA regulatory network was constructed using 4 pyroptosis-related genes, 9 miRNAs and 27 lncRNAs (Figure 8). In each lncRNA–miRNA–mRNA regulatory axis, lncRNA can enhance the expression of key pyroptosis-related genes by inhibiting corresponding miRNAs. For example, mmu-miR-298-5p can inhibit the expression of *Tnf*, thereby inhibiting inflammatory activation, and Miat can indirectly upregulate the expression of *Nlrp3* by inhibiting mmu-miR-7a-5p, thereby enhancing the inflammatory response. Overall, the ceRNA network provides a reference for understanding potential post-transcriptional regulatory mechanisms and selecting noncoding RNA treatment agents for pyroptosis-related genes with key differences.

### 2.6. Immuno-correlation Analysis of Key Pyroptosis-Related Genes

To further explore the relationship between pyroptosis-related genes and the immune microenvironment, we used the CIBERSORTx algorithm to assess the infiltration abundance of immune cell subpopulations. We found significantly elevated levels of M1 macrophage and plasma cell infiltration in the lungs of *C. muridarum*-infected mice (Figure 9A). Correlation analysis can provide key clues for studying the function and mechanism of genes related to pyroptosis. Therefore, the correlation between the expression level of eight pyroptosis-related genes and the infiltration level of immune cells was evaluated. As shown in Figure 9B, the expression of the eight key pyroptosis-related genes was strongly positively correlated with plasma cells, M1 macrophages and activated DC.

## 3. Discussion

Until now, most studies on *Chlamydia trachomatis* have focused on conjunctival and reproductive tract infections [18,19], but *Chlamydia trachomatis* also frequently causes respiratory infections clinically. Chlamydia infection is now generally recognized as a chronic inflammatory disease [20], and our previous studies also found elevated levels of inflammatory cytokines and chemokines in the lung tissue of *C. muridarum*-infected mice [21], suggesting that inflammation plays a crucial role in chlamydia respiratory infection. Pyroptosis is a kind of inflammatory cell death characterized by the release of many proinflammatory factors and is closely related to the pathogenesis of various chronic inflammatory diseases [7,10,22]. The molecular mechanism of inflammation in chlamydia respiratory tract infection is unclear and needs further study.

Bioinformatics methods have been widely used to explore key pathogenic factors and potential therapeutic targets for *Chlamydia trachomatis* reproductive tract and conjunctival infections [23,24,25]. However, to our knowledge, the bioinformatics analysis of pyroptosis-related genes in *Chlamydia trachomatis* respiratory infection has not been reported. In this study, for the first time, 13 differentially expressed pyroptosis-related genes in *C. muridarum* respiratory tract infection were identified by bioinformatics methods. At the same time, GO and KEGG analyses were performed on these 13 differentially expressed pyroptosis-related genes to clarify the enrichment relationship between these genes and related pathways. The results showed that these genes are enriched in the inflammasome complex, interleukin-1 receptor binding and other pathways, confirming the role of scorch death in chlamydia respiratory infection. Through PPI analysis, we screened out the eight most exciting genes related to pyroptosis. The results of qPCR showed that the mRNA levels of *Tnf*, *Tlr2*, *Il1b*, *Nlrp3*, *Tlr9*, *Mefv*, *Tnfaip3* and *Zbp1* were consistent with the bioinformatics analysis results. Previous studies have shown that *C. muridarum* can activate lung tissue inflammation through the TLR2/MyD88 signaling pathway [26], along with the activation of inflammatory bodies NLRP3, pro-IL-1β and TLR4 [27], which play an important role in cell pyroptosis [28]. At the same time, it has been reported that these potential pyroptosis genes promote cell pore formation by releasing inflammatory cytokines such as IL-1β and IL-18 and activating GSDMD [29,30]. Saving damaged lung tissue is the treatment strategy for *Chlamydia trachomatis* respiratory infection. Therefore, further elucidating the pyroptosis-related genes in the above lung tissues will provide a basis for further understanding the pathogenesis of lung tissue inflammation induced by respiratory infection of *Chlamydia trachomatis*.

Among the proteins encoded by these genes, TNF can enhance the expression of NLRP3 through the NF-κB signaling pathway [28], and NLRP3, as an inflammatory body, can activate caspase-1 by binding protein ASC after activation [31] and induce pyroptosis through the classical pathway [32,33]. IL-1β is a proinflammatory cytokine produced by activated macrophages and monocytes, but it must be cleaved by caspase-1 to form its mature active form [31]. In addition to the classical caspase-1 pathway and the nonclassical caspase-4/5/11 pathway [8,34], there is also a caspase-3-induced pyroptosis pathway [35]. Unlike these two pathways, caspase-3 mainly cleaves GSDME and induces pyroptosis [36]. However, we limited the effect of GSDME-mediated pyroptosis on cancer therapy. In this study, we confirmed that chlamydia infection could upregulate the expression of caspase-3 and GSDME in the lung, suggesting that caspase-3 is involved in pyroptosis in *Chlamydia trachomatis* infection, which is worthy of further exploration.

Recently, more and more studies have proven that lncRNA and miRNA-mediated pyroptosis are involved in the occurrence of inflammatory diseases. In studies on sepsis, miRNA miR-125b-5p has been found to play an active role in inflammation, directly targeting Keap1 and reducing cell-death-induced inflammation through the Keap1/Nrf2/GPX4 pathway, thereby improving acute lung injury caused by sepsis [37,38]. In intestinal ischemia–reperfusion (II/R) injury, miR-351-5p was shown to target MAPK13 and sirtuin-6, activate the Bcl2-1/NF-κB signaling pathway and cleave caspase-3, which promotes cell death [39]. In a study on *Chlamydia trachomatis* infection, it was found that lncRNA ZEB1-AS1 acts as a sponge of miR-1224-5p, targeting MAP4K4 to activate the MAPK/ERK pathway, thereby promoting cell death [40]. Therefore, we constructed the ceRNA regulatory network to understand the interrelationship between lncRNA and miRNA and their role in regulating the expression of pyroptosis-related genes in chlamydia infection. Using the multiMiR database, we predicted that nine miRNAs regulate four important pyroptosis-related genes: *Tnf*, *Il1b*, *Nlrp3* and *Tnfaip3*. Subsequently, we found possible interactions between 27 lncRNAs and these 9 miRNAs. Based on these findings, we built the ceRNA network.

Macrophages play a central role in regulating inflammatory response and can exhibit two different phenotypes under different microenvironments and stimuli [41]: M1 proinflammatory macrophages and M2 anti-inflammatory macrophages [42]. M1 macrophages mainly secrete inflammatory cytokines such as IL-1β, TNF-α and IL-18 to induce inflammation [43]. Dendritic cells are heterogeneous cells that detect the surrounding microenvironment and induce host tolerance or trigger host defense proinflammatory responses [44]. Mature activated DC shows upregulation of proinflammatory cytokine levels, aggravating the body’s inflammatory response [45]. Studies on sepsis have shown that TNF-α upregulates the expression of NLRP3 through the NF-κB signaling pathway, induces polarization of macrophages to M1 macrophages and causes pyroptosis of lung macrophages, leading to histological lesions [46]. In this study, eight pyroptosis-related genes identified were significantly positively correlated with M1 macrophages and activated DC, suggesting that the occurrence of pyroptosis in *Chlamydia trachomatis* infection may be related to the polarization of M1 macrophages and the activation of DC, which provides a new understanding of pyroptosis in *Chlamydia trachomatis* infection.

## 4. Materials and Methods

### 4.1. Mice

Female C57BL/6 mice aged 6–8 weeks (weighing 18–20 g) were used for all experiments. All mice were from Huafukang Biotechnology Co., Ltd. (Beijing, China). The mice were reared under specific pathogen-free conditions at Tianjin Medical University. All animal procedures were reviewed and approved by the Animal Ethical and Welfare Committee (AEWC) of Tianjin Medical University (number of animal permit: SYXK: 2016-0012; approval date: 7 March 2018).

### 4.2. C. muridarum Respiratory Tract Infection Models

The *Chlamydia trachomatis* mouse pneumonitis biovar, also known as *C. muridarum*, was presented by Dr. Xi Yang (The University of Manitoba, Canada). *C. muridarum* culture, purification and quantification were performed as previously described [47]. After anesthetizing mice with isoflurane, a 40 μL sucrose phosphate glutamic acid (SPG) buffer of *C. muridarum* containing 1 × 10^3^ inclusion forming units (IFU) was used to inoculate mice intranasally.

### 4.3. RNA Extraction from Lung Tissues and Sample Preparation for Transcriptome Sequencing

The total RNA was extracted according to the instruction manual for TRIzol Reagent (Life Technologies, Carlsbad, CA, USA). In 8 individual samples (4 from the lungs of the uninfected C57BL/6 mice and 4 from the lungs of C57BL/6 infected with *C. muridarum* for seven days), RNA concentration and purity were measured using NanoDrop 2000 (Thermo Fisher Scientific, Wilmington, DE, USA). RNA integrity was assessed using the RNA Nano 6000 Assay Kit of the Agilent Bioanalyzer 2100 system (Agilent Technologies, Santa Clara, CA, USA). RNA sequencing assay was performed and analyzed by Biomarker Technologies Corporation (Beijing, China). Raw data (raw reads) of FASTQ format were first processed through in-house Perl scripts. Raw sequences were transformed into clean reads after data processing. These clean reads were then mapped to the reference genome sequence. Only reads with a perfect match or one mismatch were further analyzed and annotated based on the reference genome. Hisat2 tools were used to map to the reference genome.

### 4.4. Principal Component Analysis

Pearson’s correlation test verified intragroup data repeatability in each group. The intragroup data repeatability of the dataset was tested by sample clustering analysis. Statistical analysis was performed with R software (version 4.1.3), and the “ggplot2” package presented the results.

### 4.5. Identifying Differently Expressed Pyroptosis-Related Genes

We normalized and preprocessed the data, then screened differentially expressed genes between *C. muridarum*-infected samples and control groups via the “limma” package in R language. Genes with |log_2_ fold changes| > 2 and *p* < 0.05 were considered differentially expressed genes. The search for pyroptosis-related genes was performed in the GEO database (http://www.ncbi.nlm.nih.gov/geo, accessed on 5 December 2022), and 184 pyroptosis-related genes were obtained. Overlapping genes between differentially expressed genes and pyroptosis-related genes were identified as differently expressed pyroptosis-related genes and visualized by the “ggvenn” package of R software (version 4.1.3). The correlation analysis of differently expressed pyroptosis-related genes used Spearman correlation in the “corrplot” package.

### 4.6. Function and Pathway Enrichment Analysis

We used the “clusterProfiler” package of R software (version 4.1.3) to perform the GO and KEGG enrichment analyses of differently expressed pyroptosis-related genes. GO analysis included three categories, biological process (BP), cellular component (CC) and molecular function (MF), which was influential in the exploration of biological functions. KEGG analysis was used to explore potential pathways. The results were visualized by the “enrichplot” and “ggplot2” packages.

### 4.7. Construction of PPI Network and Module Analyses

The STRING database (https://string-db.org/, accessed on 14 February 2023) was used to construct the PPI network of DEG-encoded proteins [19]; the threshold was set to a combined score ≥ 0.4, and the file in tsv format was downloaded. Then, Cytoscape software (version 3.9.1) was used to visualize the PPI network, and cytoHubba was used to excavate the eight hub genes.

### 4.8. qPCR

Total RNA was isolated from lung tissue of *C. muridarum*-infected and healthy mice with TRIzol Reagent (Invitrogen, Carlsbad, CA, USA) and purified using isopropanol, 75% ethanol and RNase-free water following the manufacturer’s instructions. After RNA concentration and purity were determined, cDNA was synthesized using One-Step gDNA Removal and cDNA Synthesis SuperMix (TransGen, Beijing, China). The qPCR was performed using the 2xRealStar Fast SYBR qPCR Mix (GenStar, Beijing, China) and proceeded on LightCycler96 (Roche, Basel, Switzerland). With β-actin as the internal reference gene, the mRNAs expression level of the target genes was quantized by 2^−ΔΔCt^ mode. Primer sequences were shown in Appendix A.

### 4.9. Western Blot Analysis

*C. muridarum*-infected mice lung tissues were lysed with RIPA lysis buffer containing protease and phosphatase inhibitors, and protein concentrations were determined using a BCA (Thermo Fisher Scientific, Rockford, IL, USA) kit. Subsequently, the membrane was placed in a 5% blocking buffer with bovine serum albumin for 1 h, incubated with a primary antibody against caspase-3 (Proteintech, Chicago, IL, USA), GSDMD (Cell Signaling Technology, Danvers, MA, USA), GSDME (Cell Signaling Technology, Danvers, MA, USA) and β-actin (Abcolonal, Shanghai, China) at 4 °C overnight, and then incubated with horseradish peroxidase-labeled secondary antibodies (Absin, Shanghai, China) at room temperature for 1 h. Finally, an enhanced chemiluminescence kit was used for visualization. The collected images were examined with ImageJ software (version 6.0; Media Cybernetics, Inc., Rockville, MD, USA).

### 4.10. Construction of the ceRNA-Regulating Network

The “multiMiR” package in R software (version 4.1.3) combines 14 databases, including the mirTarbase database (https://maayanlab.cloud/Harmonizome/resource/MiRTarBase, accessed on 11 May 2023), with experimental methods to verify the relationship. It uses this tool to predict the miRNAs of the pyroptosis-related genes of interest. All the lncRNAs–miRNAs interaction data were obtained in the Starbase database (https://starbase.sysu.edu.cn/, accessed on 23 May 2023), and the target lncRNAs were screened according to clipExpNum >7. Finally, visualization was carried out in Cytoscape software.

### 4.11. Analysis of Immune Cell Infiltration

We used the CIBERSORTx algorithm in R software (version 4.1.3) to obtain an immune infiltration matrix from RNA-seq data. The “ggplot2” package was used to visualize each sample and group’s immune infiltration matrix data. T-test was used to compare the difference between the two groups. We then used the “pheatmap” package in R software (version 4.1.3) to create correlated heatmaps, visualizing the correlations between key genes and immune cells.

### 4.12. Statistical Analysis

Data analyses and statistical tests were performed in RStudio (version 4.1.3). Each in vitro experiment was repeated at least three times. Statistical analysis was performed using GraphPad Prism software (version 9.0). Based on the homogeneity of the variance test, Student’s t-test was performed, and *p* < 0.05 was considered significant and statistically significant.

## Figures and Tables

**Figure 1 ijms-24-13570-f001:**
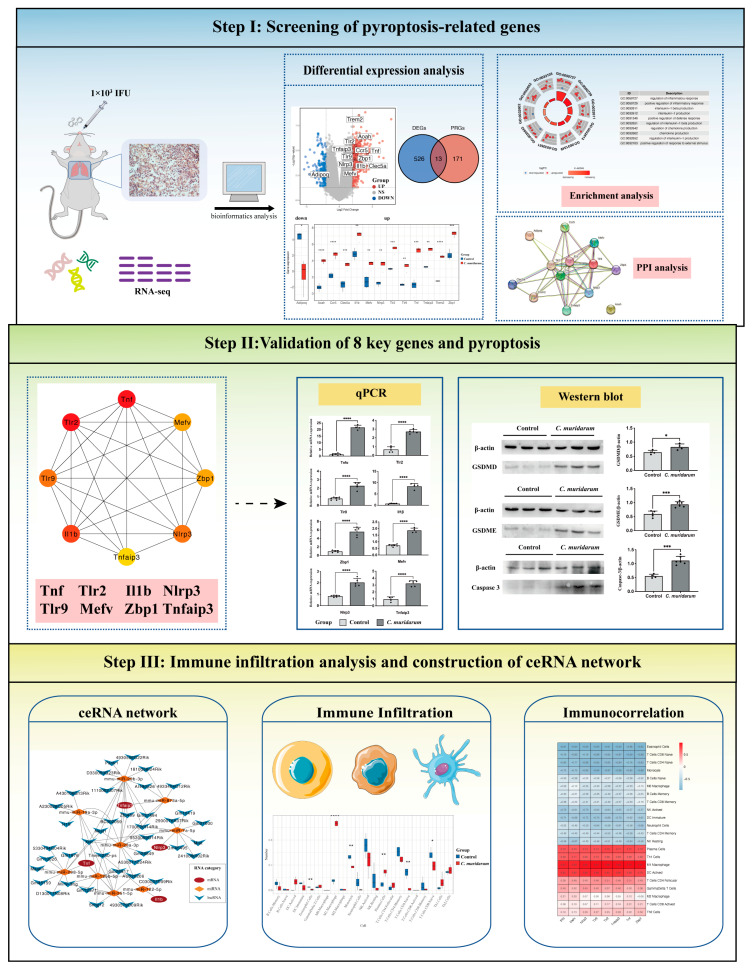
The design idea of this study. A mouse infection model of *C. muridarum* was constructed, and transcriptome sequencing was performed. Data were processed in R software (version 4.1.3), using tools such as quality control, normalization and background correction. Thirteen DEGs were identified by the differential analysis, and GO and KEGG enrichment analyses were performed. At the same time, eight hub genes were identified by PPI analysis and then verified by qPCR. Finally, the ceRNA regulatory network was constructed through the multiMiR and starbase databases to reveal the specific mechanism of PRGs in *C. muridarum* infection. * *p* < 0.05, ** *p* < 0.01, *** *p* < 0.001, **** *p* < 0.0001. Abbreviations: *C. muridarum*, *Chlamydia muridarum*; DEGs, differentially expressed genes; GO, gene ontology; KEGG, Kyoto Encyclopedia of Genes and Genomes; PPI, protein–protein interaction; qPCR, quantitative real-time PCR; ceRNA, competitive endogenous RNA; PRGs, proptosis-related genes.

**Figure 2 ijms-24-13570-f002:**
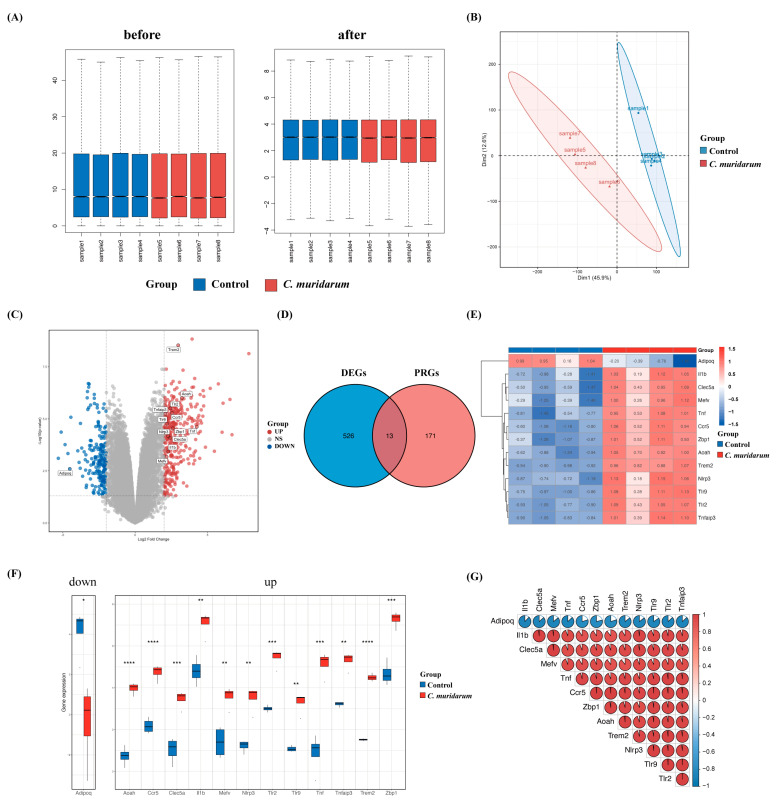
Screening of differentially expressed pyroptosis-related genes. (**A**) Data normalization. The box plots shows each sample’s gene expression level before and after normalization. The blue bars represent healthy mice, denoted by “Control”, and the red bars represent *C. muridarum*-infected mice, denoted by “*C. muridarum*”. (**B**) Principal component analysis. (**C**) Volcano map. It contains 358 significantly upregulated genes, which are represented by red dots, and 81 significantly downregulated genes, represented by blue dots. The gray dots represent stably expressed genes. (**D**) Venn plot. The blue circle represents DEGs, and the red circle represents PRGs; the intersection represents differentially expressed PRGs. (**E**) Thirteen heatmaps of differentially expressed PRGs. It contains 1 significantly downregulated gene and 12 significantly upregulated genes. (**F**) Box plot of 13 differentially expressed PRGs. The blue bar represents healthy mice, denoted by “Control”, and the red bar represents *C. muridarum*-infected mice, denoted by “*C. muridarum*”. (**G**) Correlation analysis of 13 differentially expressed PRGs. * *p* < 0.05, ** *p* < 0.01, *** *p* < 0.001, **** *p* < 0.0001. Abbreviations: *C. muridarum*, *Chlamydia muridarum*; DEGs, differentially expressed genes; PRGs, proptosis-related genes.

**Figure 3 ijms-24-13570-f003:**
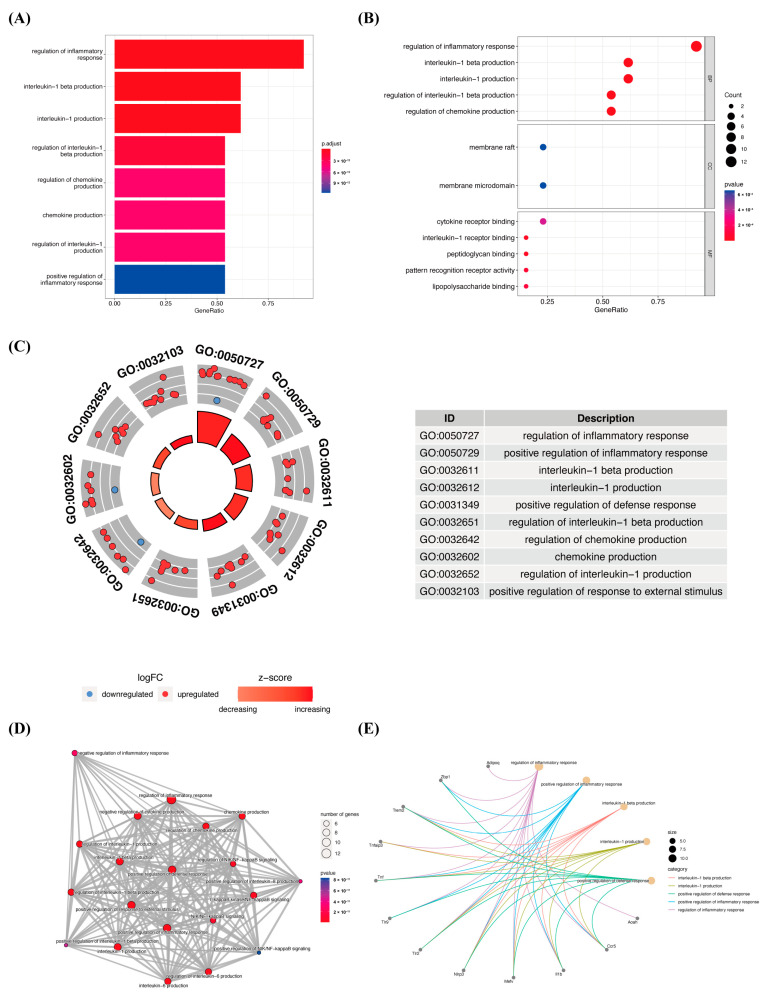
GO enrichment analysis of 13 differentially expressed PRGs. (**A**) Bar plot of enriched GO terms. (**B**) Bubble plot of enriched GO terms. (**C**) Ten diagrams of enriched GO terms. It contains three aspects—BP, CC and MF—and shows the genes involved in each GO term. (**D**) Relationship between the pathways obtained by GO enrichment analysis. (**E**) Common genes among the three most prominent pathways. Lines of different colors represent each path. Abbreviations: GO, gene ontology; PRGs, proptosis-related genes.

**Figure 4 ijms-24-13570-f004:**
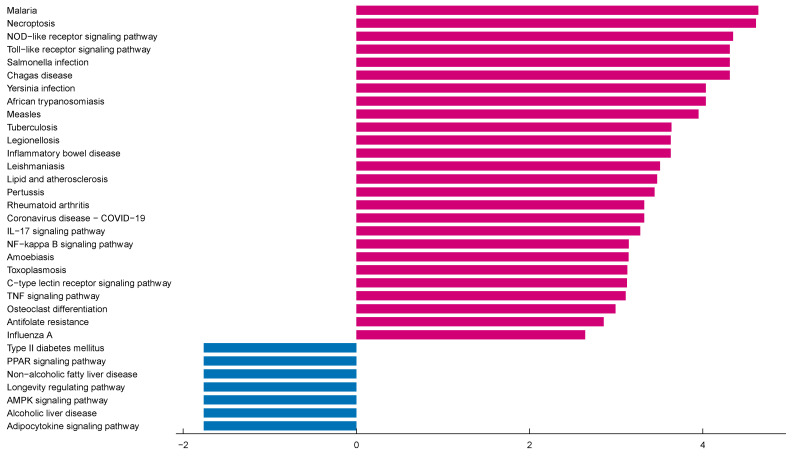
KEGG pathway analysis of 13 differentially expressed PRGs. According to the adjusted *p*-value, 55 pathways were reported. Abbreviations: KEGG, Kyoto Encyclopedia of Genes and Genomes; PRGs, proptosis-related genes.

**Figure 5 ijms-24-13570-f005:**
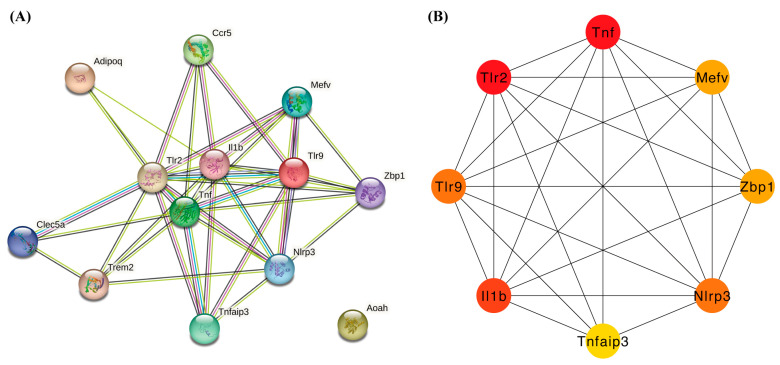
Construction of PPI network and identification of hub genes. (**A**) The PPI network of 13 differentially expressed PRGs was constructed using the STRING database. It contains 13 nodes and 40 edges. The average node degree is 6.15, and the PPI enrichment *p*-value is less than 1.0 × 10^−16^. (**B**) First eight hub genes of the PPI network. The cytoHubba plug-in of Cytoscape identified the first eight genes with the highest degree. These genes are ranked in descending order from red to yellow. Abbreviations: PPI, protein–protein interaction; PRGs, proptosis-related genes.

**Figure 6 ijms-24-13570-f006:**
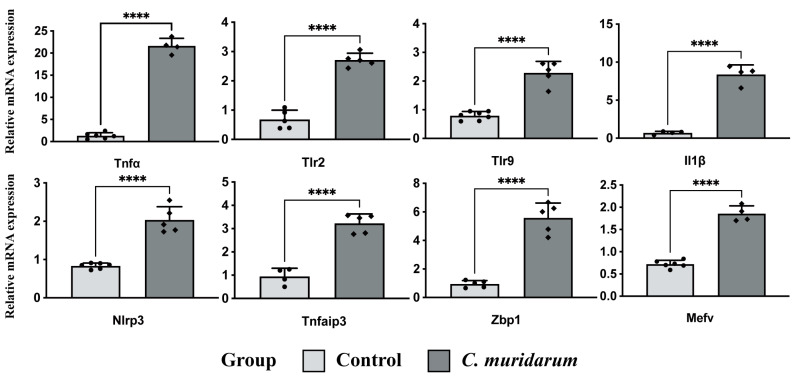
qPCR experiments to verify the expression of PRGs. Data are represented as means ± SD from n = 3–4 per group, representing one of three independent experiments. A two-sided unpaired Student’s *t*-test determines the statistical significance of differences. **** *p* < 0.0001. Abbreviations: qPCR, quantitative real-time PCR; PRGs, proptosis-related genes; *C. muridarum*, *Chlamydia muridarum*.

**Figure 7 ijms-24-13570-f007:**
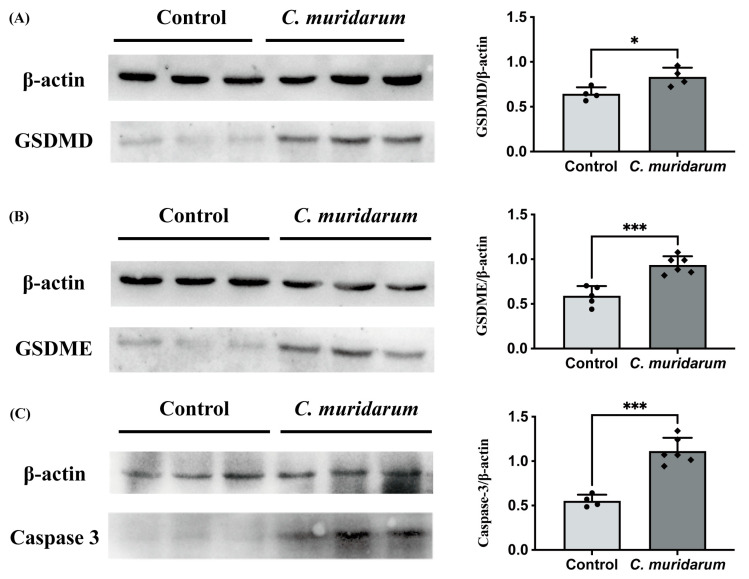
Western blot of caspase-3 and gasdermin proteins in the *C. muridarum*-infected model. (**A**) GSDMD (53 kDa) was examined. β-actin (42 kDa) was used as a control. (**B**) GSDME (57 kDa) was examined. β-actin (42 kDa) was used as a control. (**C**) Caspase-3 (34 kDa) was examined. β-actin (42 kDa) was used as a control. Data are expressed as mean ± SD, with n = 3–4 for each group, representing one of three independent experiments. A two-sided unpaired Student’s t-test determines the statistical significance of differences. * *p* < 0.05, *** *p* < 0.001.

**Figure 8 ijms-24-13570-f008:**
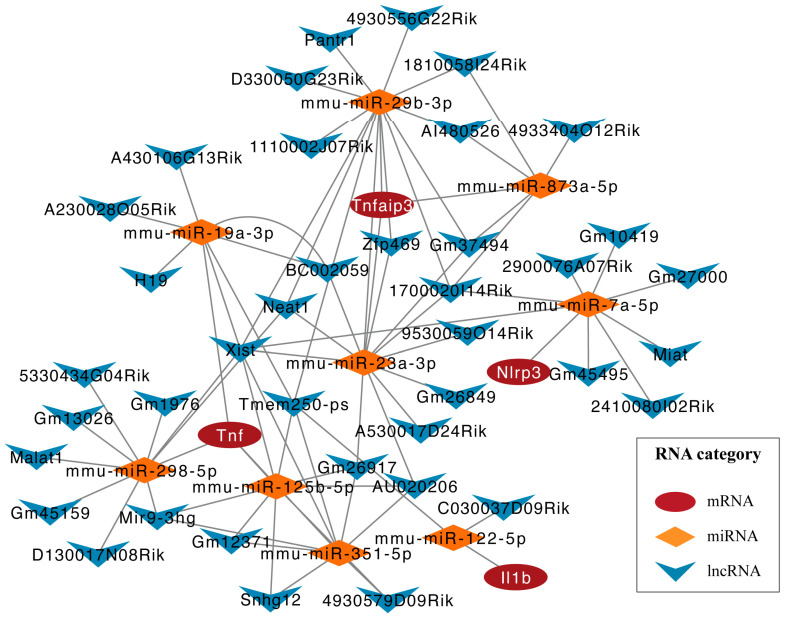
ceRNA-regulating networks. The red ovals represent protein-coding genes, the orange diamonds represent miRNAs, and the blue V represents lncRNAs. The black line indicates the lncRNA–miRNA–mRNA interaction. Abbreviations: *C. muridarum*, *Chlamydia muridarum*; ceRNA, competing endogenous RNA; mmu, mus musculus; miRNAs, microRNAs; lncRNA, long noncoding RNAs.

**Figure 9 ijms-24-13570-f009:**
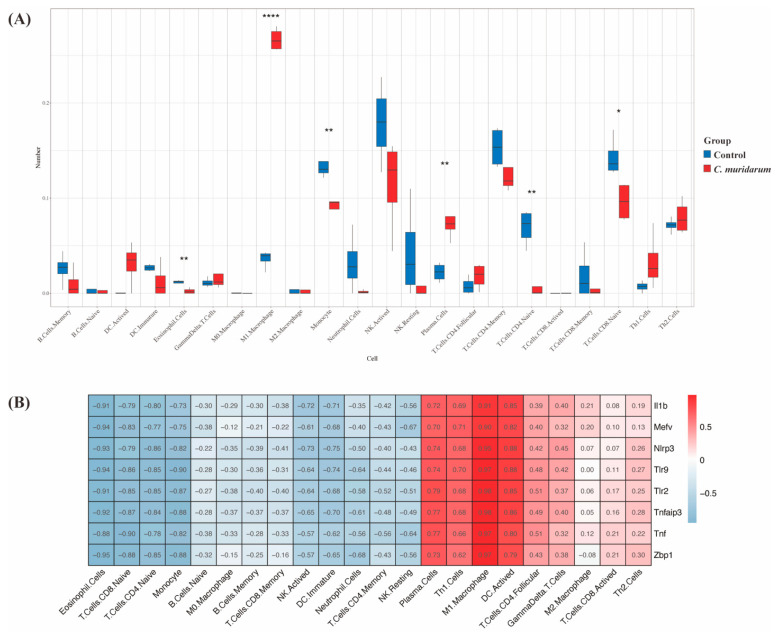
Immune microenvironment and immuno-correlation analysis of differently expressed PRGs. (**A**) Differences in the abundance of 22 immune cell types. The blue bars represent healthy mice, denoted by “Control”, and the red bars represent *C. muridarum*-infected mice, denoted by “*C. muridarum*”. The horizontal axis shows the 22 immune cell types, and the vertical axis shows the abundance of immune cell infiltration. (**B**) Correlation analysis between eight differently expressed PRGs and immune infiltration. The redder the color, the stronger the correlation. The bluer the color, the weaker the correlation. * *p* < 0.05, ** *p* < 0.01, **** *p* < 0.0001. Abbreviations: PRGs, pyroptosis-related genes; *C. muridarum*, *Chlamydia muridarum*.

**Table 1 ijms-24-13570-t001:** Top 7 GO terms and 3 KEGG pathway analyses for the differentially expressed PRGs.

Pathway ID	Category	Description	Count	Genes	*p*-Value
GO:0061702	CC	inflammasome complex	2	*Mefv*, *Nlrp3*	4.32 × 10^−5^
GO:0005149	MF	interleukin-1 receptor binding	2	*Il1b*, *Tlr9*	2.31 × 10^−5^
GO:0050727	BP	regulation of inflammatory response	12	*Aoah, Ccr5, Il1b, Mefv, Nlrp3*, *Tlr2, Tlr9, Tnf, Tnfaip3, Trem2, Zbp1, Adipoq*	1.86 × 10^−22^
GO:0050729	BP	positive regulation of inflammatory response	9	*Ccr5, Il1b, Mefv, Nlrp3, Tlr2, Tlr9, Tnf, Trem2, Zbp1*	1.31 × 10^−18^
GO:0032611	BP	interleukin-1 beta production	8	*Ccr5, Il1b, Mefv, Nlrp3, Tlr2, Tnf, Tnfaip3, Trem2*	2.28 × 10^−17^
GO:0031349	BP	positive regulation of defense response	9	*Ccr5, Il1b, Mefv, Nlrp3, Tlr2, Tlr9, Tnf, Trem2, Zbp1*	9.81 × 10^−16^
GO:0032675	BP	regulation of interleukin-6 production	7	*Ccr5, Il1b, Tlr2, Tlr9, Tnf, Tnfaip3, Trem2*	3.13 × 10^−13^
mmu04217	KEGG	necroptosis	5	*Il1b, Nlrp3, Tnf, Tnfaip3, Zbp1*	6.17 × 10^−7^
mmu04621	KEGG	NOD-like receptor signaling pathway	5	*Il1b, Mefv, Nlrp3, Tnf, Tnfaip3*	1.70 × 10^−6^
mmu04620	KEGG	Toll-like receptor signaling pathway	4	*Il1b, Tlr2, Tlr9, Tnf*	2.82 × 10^−6^

Abbreviations: GO, gene ontology; CC, cellular components; BP, biological processes; MF, molecular functions; KEGG, Kyoto Encyclopedia of Genes and Genomes; PRGs, proptosis-related genes.

**Table 2 ijms-24-13570-t002:** PPI network of the top 8 hub genes.

Rank	Gene Symbol	Score	Function	Gene Name
1	*Tnf*	900	Pyroptosis core components	Tumor necrosis factor
1	*Tlr2*	900	Inflammasome	Toll-like receptor 2
3	*Il1b*	894	Pyroptosis core components	Interleukin-1 beta
4	*Nlrp3*	864	Inflammasome	NLR family pyrin domain containing 3
4	*Tlr9*	864	Inflammasome	Toll-like receptor 9
6	*Mefv*	720	Inflammasome	Mediterranean fever
6	*Zbp1*	720	Genes with pyroptosis function or regulate the expression of PRGs	Z-DNA binding protein 1
8	*Tnfaip3*	120	Genes with pyroptosis function or regulate the expression of PRGs	Tumor necrosis factor, alpha-induced protein 3

Abbreviations: PPI, protein–protein interaction.

**Table 3 ijms-24-13570-t003:** miRNAs and specifically targeted mRNAs in the ceRNA regulatory network.

mRNA	miRNA
*Tnf*	mmu-miR-125b-5p, mmu-miR-351-5p, mmu-miR-298-5p, mmu-miR-19a-3p
*Il1b*	mmu-miR-122-5p
*Nlrp3*	mmu-miR-7a-5p
*Tnfaip3*	mmu-miR-29b-3p, mmu-miR-873a-5p, mmu-miR-23a-3p

Abbreviations: miRNA, microRNAs; ceRNA, competing endogenous RNA.

## Data Availability

The raw data used to support the findings of this study are available from the corresponding author upon request.

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
