# Peer review of "In Silico Identification and Validation of Pyroptosis-Related Genes in Chlamydia Respiratory Infection"

_ijms, 2023, doi:10.3390/ijms241713570_

Round 1

Reviewer 1 Report

This research aims to discover and build a competitive network of endogenous RNA (ceRNA) involved in pyroptosis during Chlamydia lung infection in a murine model. 

Although this study does not aim to identify the survival and replication of Chlamydia, it would have been interesting to know what happens with C. muridarum during the expression of these pyroptosis genes.

Because some bacterium intracellular can survive and multiply inside immune cells such as macrophages. For example, Burkholderia pseudomallei is an intracellular bacteria that is reduced growth in TLR9-depleted RAW264.7 cells infected with B. pseudomallei, suggesting that TLR9 is involved in intracellular bacterial killing in macrophages through negatively regulating cytokine production, particularly IFN-b, a vital cytokine to control pyroptosis via caspase-11 activation. Moreover, TLR9-deficient mice have also been shown to reduce the clearance of Streptococcus pneumoniae from the lungs, implying that TLR9 may also involve in bacterial clearance. Several reports have previously demonstrated that pyroptosis is essential for restricting intracellular bacterial killing. What happens with Chlamydia growth? Chlamydia no controló la apoptosis?

Zhan R, Han Q, Zhang C, Tian Z, Zhang J. 2015. Toll-Like receptor 2 (TLR2) and TLR9 play opposing roles in host innate immunity against Salmonella enterica serovar Typhimurium infection. Infect Immun 83:1641–1649. 

Albiger B, Dahlberg S, Sandgren A, Wartha F, Beiter K, Katsuragi H, Akira S, Normark S, Henriques-Normark B. 2007. Toll-like receptor 9 acts at an early stage in host defence against pneumococcal infection. Cell Microbiol 9:633–644. 

Pudla M, Sanongkiet S, Ekchariyawat P, Luangjindarat C, Ponpuak M, Utaisincharoen P. TLR9 Negatively Regulates Intracellular Bacterial Killing by Pyroptosis in Burkholderia pseudomallei-Infected Mouse Macrophage Cell Line (Raw264.7). Microbiol Spectr. 2022 Oct 26;10(5):e0348822.

Minor fixes are needed. For example, in several graphs, on the ordinate axis, they do not comment on which units the results are being evaluated. Several acronyms do not describe their meaning. The method for growth and purification of Chlamydia from the cell line is not well described. For this, the authors suggest a reference that does not have an adequate description.

English is adequate in this version and only requires minor corrections and some editing, but this would be minimal.

Reviewer 2 Report

Pyroptosis is an inflammation-related cell death program, that plays crucial roles in host defense against infections through the release of proinflammatory cytokines and cell lysis. In this study, Sun R et al., identified and validated genes involved in pyroptosis using Chlamydia muridarum, mouse model for Chlamydia trachomatis. Overall, the study is well performed, and the results evidently demonstrate the conclusion. However, the reviewer advises a comprehensive review of the manuscript for proper English usage. In addition, authors are suggested changing the title to ‘in silico identification………infection’.

Extensive editing of english language required.

Authors are suggested to get the manuscript checked for proper English usage.

Reviewer 3 Report

The paper titled “ Identification and validation of pyroptosis-related genes in chlamydia respiratory infection based on bioinformatics” (ijms 2569718) describes the role of pyroptosis in the occurrence and development of Chlamydia trachomatis respiratory infection. The Authors found the significantly reduced expression of analyzed genes (Tnf, Tlr2, Il1b, Nlrp3, Tlr9, Mefv, Zbp1, and Tnfaip3) in the lungs of C. muridarum-infected mice. The manuscript is very interesting, but some issues require clarification.

1. Methods should be described before the discussion

2. C. trachomatis infects humans and C. muridarum naturally infects only members of the family Muridae. What is the clinical meaning of your study (e.g., anti-inflammatory effect or cancer therapy)? Is this research model compatible with the involvement of genetic factors associated with chlamydia infections in humans?

3. Which of the analysed genes (Tnf, Tlr2, Il1b, Nlrp3, Tlr9, Mefv, Zbp1 and Tnfaip3) might have the strongest influence on pyroptosis? Can we tell about some predisposition to a pro-inflammatory state or pyroptosis?

4. Pyroptosis is observed in the immune system, the nervous system, and the cardiovascular system. What symptoms are observed during excessive pyroptosis in animals and humans? Do they differ?
